# Actuarial Patency Rates of Hepatico-Jejunal Anastomosis after Repair of Bile Duct Injury at a Reference Center

**DOI:** 10.3390/jcm11123396

**Published:** 2022-06-13

**Authors:** Włodzimierz Otto, Janusz Sierdziński, Justyna Smaga, Oskar Kornasiewicz, Krzysztof Dudek, Krzysztof Zieniewicz

**Affiliations:** 1Department of General, Transplant & Liver Surgery, University Medical Center, Medical University of Warsaw, Banacha 1a, 02-097 Warsaw, Poland; wotto@wum.edu.pl (W.O.); jsmaga@o2.pl (J.S.); oskar.kornasiewicz@wum.edu.pl (O.K.); krzysztof.dudek@wum.edu.pl (K.D.); krzysztof.zieniewicz@wum.edu.pl (K.Z.); 2Department of Medical Informatics and Telemedicine, Medical University of Warsaw, Litewska 14/16, 00-581 Warsaw, Poland

**Keywords:** bile duct injury, actuarial patency rate after bile duct repair, Strasberg classification of bile duct injury

## Abstract

Background: Bile duct injury complicates patients’ lives, despite the subsequent repair. Repairing the injury must restore continuity of the bile tree and bring the patient into a state of cure referred to as “patency”. Actuarial primary or actuarial secondary patency rates, depending on whether the patient underwent primary or secondary repair of injury, are proposed to be a proper metric in evaluating outcomes. This study was undertaken to assess outcomes of 669 patients with bile duct injuries Strasberg D and E type referred to the department from public surgical wards between 1990 and 2020. In 442 patients, no attempt was made to repair prior to a referral, and in 227 an attempt to repair was made which failed. Methods: Observations were summarized on December 31st, 2020. The retrospective analysis included: primary patency attained (Grade A result), secondary patency attained (Grade C result), patency loss, and actuarial patency rates of the bile tree at 2, 5, and 10 years. Results: Twenty-five (3.7%) patients died after repair surgery. Actuarial patency rates at 2, 5, and 10 years of follow-up were 93%, 88%, and 74% or 86%, 75%, and 55% in patients attaining Grade A and Grade C outcomes, respectively (*p* < 0.001). Conclusion: Bile duct injury stands out as a surgical challenge, requiring specialized management at a referral center. Improper proceeding after an injury is the factor leading to faster loss of anastomotic patency.

## 1. Introduction

The incidence of iatrogenic main bile duct lesions has significantly increased, with incidences ranging from 0.2 to 1.5% according to current studies [1]. An incidence of bile duct injuries in the pediatric population is slightly higher than in adults probably due to less experience of pediatric surgeons with laparoscopic cholecystectomy [2]. Moreover, the incidence of CBD injuries is greater in regional hospitals than in university hospitals or larger centers [3]. The incidence of serious iatrogenic injuries to the bile tract (Strasberg C, D, E) is estimated in Poland at 0.4–0.6% of circa 50,000 laparoscopic cholecystectomies that are performed every year. Repairing this type of damage is challenging for a surgeon but for the patients, it is a hard experience that changes life into a string of unwanted health problems. The cause is a stricture developing at the site of the bile duct injury or at the site of the hepatico-jejunal anastomosis performed in the process of repair as time passes from the reconstructive surgery [4,5,6]. Late diagnosis of injury, failures in attempts to repair, as well as numerous surgical and non-surgical interventions preceding successful repair, results in the extension of injury and often increases the complexity of bile duct repair [7,8,9,10,11,12]. Most of the patients that are referred to the tertiary HPB units suffer abdominal infections because of the reasons mentioned above. These patients require a lot of additional treatment interventions, lavage, and drainage of the peritoneal cavity, as well as an intensive general supportive treatment and antibiotic therapy before the final repair operation [13,14,15,16,17]. Given the circumstances, getting a good result is usually difficult. However, an achievement of a long asymptomatic period in the patient’s life was always recognized as the most important measure of repair success, and widely discussed in many reports [18,19].

Recently, an international group of specialists in surgery and gastroenterology proposed a new standard to report outcomes after bile duct repair. The method is based on the concept of “patency” defined as an open and functionally effective bile tree. According to standard practice and literature, a period of 90 days was established for achieving primary patency after surgery. What is evaluated is how the “patency” of the bile tree is attained and conserved after treatment and, if it is lost, how effectively it is restored. The most important metric in evaluating outcome is the actuarial primary patency rate calculated as a Kaplan–Meier survival curve [20,21,22].

The study was undertaken to analyze the results of repair of the major bile duct injuries in 669 patients who were treated by surgery during the period from 1990 to 2015 and were followed up to 2020, at a specialized HPB center which is our Institution, according to the rules set out in this proposition.

## 2. Materials and Methods

### 2.1. General Considerations

The Department of General, Transplant and Liver Surgery, Medical University of Warsaw is one of the few tertiary reference centers in Poland that deal with surgical treatment of the consequences of the bile ducts disability. The analysis presented in the paper concerns a cohort of 669 patients (501 females, mean age 49, +/−14.4 years and 168 males, mean age 54.3, +/−13.3 years), with bile duct injuries sustained during laparoscopic cholecystectomy in public surgical wards. Patients were referred to the Department between 1990–2015 for advanced treatment and repair.

Inclusion criteria for the study included patients with iatrogenic bile duct injury classified as Strasberg D and E Type, sustained in adult patients during cholecystectomy in the surgical wards of public hospitals.

Exclusion criteria included patients under 18 years old, patients with iatrogenic bile duct injuries sustained during cholecystectomy performed in surgical departments of University or Third Level Referential Hospitals, patients with iatrogenic bile duct injuries other than Strasberg D and E Type, patients with injury to the vessels of hepato-duodenal ligament concomitant to the bile duct injury, patients with the acquired strictures of the bilo-jejunal anastomosis due to formerly performed repair, patients who could attain primary or secondary patency of anastomosis by endoscopic stenting.

Grouping of the patients has been established according to the history of the clinical course after the injury. Two groups of patients were analyzed. Patients who were transferred with no attempts to repair the injury by public surgeons were selected to the GROUP I, forming subgroups I a, I b, I c, according to the clinical differences. Patients who had undergone attempts to repair in public surgical wards that failed were selected for the GROUP II, forming subgroups II a, and IIb, according to the clinical differences. Indication of clinical details and differences not only between the main groups I and II but especially between the subgroups of patients seemed to be a key to finding why the actuarial patency rates after repair of injury are different.

### 2.2. Group I—Patients Transferred to the Department with No Attempts to Repair the Injury at Public Surgical Wards

The group was very diverse, and consisted of:137 patients with injury recognized intraoperatively. They were transferred soon after the index operation with the properly secured drainage of the peritoneal cavity (Group Ia),95 patients with injuries recognized during the postoperative period. Effective drainage of the peritoneal cavity was performed on all of them in public hospitals by percutaneous USG guided drainage provided in addition to a drain or drains left intraoperatively (Group Ib),210 patients with injuries recognized later during the postoperative period. The excessive surgical and/or endoscopic treatments of the biliary peritonitis and of the bile leak from the injured bile duct were performed in these cases (Group Ic). Detailed data are presented in Table 1.

### 2.3. Group II—Patients Transferred to the Department Because of Failure in Primary Repair of Injury at Public Surgical Wards

The group was also very diverse and consisted of:157 patients where the repair has failed. They were transferred because of bile leak from biliary-jejunal anastomosis, with intra-abdominal infection, ineffective drainage of the peritoneal cavity and sepsis. (Group IIa),70 patients where the repair has failed. The leakage at the site of anastomosis was healed within several days and the patients were released home. Soon, within a few weeks, they were admitted again to the public surgical or gastrointestinal units due to recurrent episodes of cholangitis caused by narrowing of the anastomosis. The patients were referred to the Department because of failure in antibiotic therapy and endoscopic treatment for re-repair (Group IIb). Detailed data are presented in Table 2.

### 2.4. Scope of Preparing Patients for Repair

The extent and severity of the injury were determined for all the patients by USG, CT, and MR-cholangiography which has been applied routinely since 1998. The ERCP was routinely used for either diagnostic purposes or to secure the outflow of bile to the alimentary tract in patients with external bile fistula or duct stricture in preparation for reconstructive surgery. The proper alimentation, antibiotic therapy, and cardiovascular treatment were provided if required before the repair. The patients of groups Ia and Ib did not require advanced preparation before the repair. On the other hand, the clinical condition of some group Ic patients, and especially the group IIa and IIb patients required a great deal of effort to prepare them for repair. The repair surgery was undertaken when the bile duct injury was adequately classified, sepsis and leaks were controlled, and the patient was stable and nutritionally optimized. Details are presented in Table 3.

The role of endoscopy in preparing patients with bile fistula and/or stricture of the anastomosis that were defined in Group II. Endoscopic stenting of the bile duct was applied in most of them. Aiming to provide effective bile drainage plastic stents or different types of SEMSs were inserted depending upon individual indications. However, it was not possible to achieve permanent patency of the bile tree, so the patients were classified for surgical reconstruction. Details related to the patients and treatment are presented in Table 2.

### 2.5. Scope of Repair Procedures

Both the primary and the secondary repairs were performed electively by the hepatico-jejunal anastomosis at the level of liver hilum when the conditions for the operation were considered favorable. The procedures were carried out according to well-established and generally known rules.

The one-layer end-to-side anastomosis with 5–0 absorbable suture was carried out with the jejunal loop of 40–60 cm in length for bile drainage. The 1–2 external silicone tubes to the intrahepatic tract were applied routinely via the anastomosis for cholangiography purposes and to prevent bile leak in the postoperative period. Cholangiography was performed routinely twice; first when the hepatic duct was prepared for anastomosis, and the second after the anastomosis was completed (Figure 1).

### 2.6. Follow-Up

Observations were conducted continuously. Information concerning the course after the period of hospital stay was obtained personally for each patient during regular observations in the outpatients’ department. Patients’ data were continuously collected by using the telemedical system of the WEB network and included information concerning defective cholecystectomy procedure, the extent of damage to the bile ducts and subsequent complications, as well as the modes of treatment applied before the referral to the Department and at the period needed for the patient to be prepared for primary or secondary repair. Data were summarized with follow-up to 31 December 2020.

### 2.7. Outcomes of the Study

The primary outcome of the study was to determine patients’ outcomes in each group calculated as the Kaplan–Meier curve. The plot of primary patency time for patients of Group I was started after 90 days of treatment. The actuarial patency rates at 2, 5, and 10 years were determined as percent of patients who attained and conserved primary patency of the anastomosis. The plot of secondary patency time for Group II patients was started when they achieved secondary patency after re-repair. The actuarial patency rates at 2, 5, and 10 years were determined as the percent of patients who attained and conserved secondary patency of the bile tree. in a separate paragraph in methodology.

The secondary outcome of the study was to determine factors that contribute to the loss of patency and influence the actuarial patency rate of bile duct repairs in a 30-year period (1990–2020).

### 2.8. Theory/Calculation

The new proposed standard of outcome reporting has not yet been extensively used. It was found that early referral to a specialized HPB center may be a predictor of loss of bile duct injury repair patency. It was also determined that bile duct re-repairs had lower actuarial patency rates when compared with primary repairs, and these differences were more pronounced if the comparison was made between patients with a higher and lower subsidy for care [22,23]. On the other hand, most experts in their reports agree that the management of patients who suffered major bile duct injuries stands out as a surgical challenge, requiring specialized management at a referral center. Moreover, bile duct injury repairs performed in specialized hepatobiliary centers are associated with better outcomes [24,25,26]. Two groups of patients with major bile duct injury are referred every year to our department: most of the patients are referred early after index operation with no attempts to repair the injury; a minority of patients are referred late and consist of patients for whom a repair was attempted at public surgical wards, but the repair failed. They are transferred with serious surgical complications, sepsis, or cholangitis developing despite currently performed treatment. Recently we reported on the outcome of 226 patients with major bile duct injury, reconstructed with hepaticojejunostomy. Their outcomes were examined by different criteria, mostly under postoperative clinical manifestations in accordance with the rules adopted at that time. However, comparison with other series was found difficult due to clinical disparities between studied groups [27]. As was indicated in a recently published study, standardized reporting outcomes after primary repair are applicable to re-repaired patients. We are convinced that a new standard of reporting outcomes after primary repair and secondary repair is applicable and allows us to compare reliably such different populations.

### 2.9. Ethics

According to Polish law, this cohort study has not required special ethical approval, however, the study was approved by a Bioethics Committee (AKBE/12/2018).

## 3. Data Analysis and Statistics

Continuous variables were expressed as a mean +/−SD, with sample representativeness of 95% confidence interval [CI]. Discrete variables were presented as numbers or letters, and categorical variables were adequately labeled. The Shapiro–Wilk test was used to assess the normality distributions of the study variables. The patient’s outcome in each group was calculated as a Kaplan–Meier curve. The plot of primary patency time for patients of Group I was started after the 90 days of treatment. The actuarial patency rates at 2, 5, and 10 years were determined as the percent of patients who attained and conserved primary patency of the anastomosis. The plot of secondary patency time for Group II patients was started when they achieved secondary patency after re-repair. The actuarial patency rates at 2, 5, and 10 years were determined as the percent of patients who attained and conserved secondary patency of the bile tree. The rates of patency time were compared statistically by using the Chi-square test. Multivariate analysis was performed by using logistic regression and the Cox proportional hazards models to search for factors affecting the patency of anastomosis after primary and secondary repair. The following items were adopted as the independent variables: sex and age of the patients, the extent of biliary damage, the scope of action taken in local surgical units after injury, the grades of complications overlapping the injury, and the repair attempts taken by local teams, timing of patients’ referral, referral patterns of relevant groups, the scope of supplemental treatment, and the period of treatment necessary to prepare patients for definitive repair. A *p*-value < 0.05 was adopted as statistically significant.

## 4. Results

### 4.1. Postoperative Course

The postoperative course was not complicated in 534 (79.9%) of the patients (F 432, mean age 49.7 and M 102, mean age 51.8). Complications occurred in 135 patients (20.1%); of Grade 1 (pain, leak of bile around stents or wound infection) in 44 patients, of Grade 2 (anemia requiring blood transfusion) in 31 patients, of Grade 3 (leak from the anastomosis, bile collection requiring additional drainage) in 18 patients, of Grade 4 (sepsis, pneumonia, organ insufficiency, thromboembolic disease) in 17 patients, and of Grade 5 in 25 patients, as it was determined in Clavien–Dindo scale [25]. Mortality in the entire group of 669 patients transferred for repair was 3.7%. Details are presented in Table 4.

### 4.2. Patients Who Achieved Primary Patency of Biliary

Primary patency of the biliary tree was achieved in 435 out of 442 patients of Group I. No attempts to injury repair were undertaken at the public surgical wards but drainage of the peritoneal cavity was secured effectively in all these cases. All these patients were transferred to our institution no later than after 25.4 (+/−17.1) days. The efforts and time needed to prepare the patients for repair surgery differed significantly between groups (*p* < 0.01). The clinical condition made it possible to undertake a corrective operation after a mean of 8.5 (+/−5.9) days of injury in 137 patients of Group Ia and after a mean of 15.6 (+/−7.1) days of injury in 95 patients of Group Ib. The two hundred and ten patients of Group Ic required sometimes even more than 3 weeks (mean 21.1, +/−15.4 days) of preparation for repair. Of them, seven patients died after repair in the postoperative period because of cardiovascular disorders, which was 1.59% of the population of Group I, but 1% of 665 patients who were totally transferred. The rate of attaining primary patency of biliary tree in the patients of Group I was assessed at 98.41%. Primary patency of biliary anastomosis was maintained thereafter in all these cases, so they were classified as achieving Grade A result of the repair.

Actuarial primary patency rate of patients who attained Grade A repair outcomes varied. There were no significant differences in outcome between Group Ia and Ib during the whole time of observation. The results in patients of Group Ic were significantly worse. The discrepancy appeared after the second year of follow-up (Chi-square test = 13.84, *p* < 0.001). Further observations indicated that primary patency rates at 2, 5, and 10 years of follow-up were at the same level for patients of Group Ia and Ib, but it was much lower in patients of Group Ic. The patency rates at the relevant time of observations were 98%, 91%, and 83% for patients of Group Ia, 96%, 90%, and 75% for patients of Group Ib, and 87%, 82%, and 60% for patients of Group Ic. The difference in the outcome between patients of Group Ia and Ib, and patients of Group Ic, at 10 years of follow-up was statistically significant (Chi-square test = 4.71, *p* < 0.003). The curves of actuarial patency rates and numbers of patients at risk are presented in Figure 2.

### 4.3. Patients Who Did Not Achieve Primary Patency of Biliary Tree

Primary patency of bile tree was not achieved in 157 patients of Group IIa, and it was lost in 70 patients of Group IIb. All these patients were transferred to our unit with significant delay due to failure in attempts to repair the injury in public surgical wards. The efforts and time needed to prepare these patients for re-repair also differed significantly between groups (*p* < 0.003). The clinical condition made it possible to undertake a corrective operation after a mean of 30.8 (+/−22.1) days in 157 Group IIa patients and after a mean of 18.3 (+/−17.4) days in 70 Group IIb patients. Secondary patency was achieved and maintained by a second surgical bile duct reconstruction in all of these patients, except for 18 patients of Group IIa who died due to postoperative complications, which was 7.92% of the population from Group II, but 2.7% of 665 patients who were totally transferred. The rate of attaining secondary patency of biliary tree in the patients of Group II was assessed at 92.08%. Secondary patency of the biliary anastomosis was maintained thereafter in all these cases, so they were classified as achieving a Grade C repair result.

Actuarial secondary patency rates in 139 patients of Group IIa and in 70 patients of Group IIb who attained a Grade C outcome of repair were similar. Losing patency in both groups was greatest at 2 years and after 5–7 years of observations. Secondary patency rates of anastomosis at 2, 5, and 10 years of follow-up were 86%, 83%, and 53% in patients of Group IIa, and 87%, 84%, and 55% in patients of Group IIb. The outcome at 10 years of observation in patients of Group IIa and IIb were very similar (Chi-square test = 0.33, *p* = 0.74). The curves of secondary patency rates and numbers of patients at risk are presented in Figure 3.

### 4.4. Clinical Differences between Patients Achieving Grade A and C Result of Repair

Clinical differences related to the timing of referral and to the timing of repair, in relation to results of repair in studied groups, were significant. Details are presented in Table 5.

### 4.5. Late Results in Patients with Grade A and Grade C Result of Repair

To compare the late result between 435 patients who attained repair of Grade A and 209 patients who attained repair of Grade C, the Kaplan–Meier curve for patients of Group Ia, Ib, and Ic together and patients of Group IIa and IIb together, was plotted as if it were the first repair for each of the patient in these groups. Calculation showed significantly worse outcomes in patients attaining the result of Grade C compared to patients attaining the result of Grade A. The greatest loss of patency rates in patients attaining Grade C outcomes appeared at 2 years (Chi-square test = 13.84, *p* < 0.001) and at 5–7 years of observations (Chi-square test = 11.04, *p* < 0.003). Actuarial patency rates of anastomosis at 2, 5, and 10 years of follow-up were 93%, 88%, and 74% in patients attaining a Grade A outcome and 86%, 75%, and 55% in patients attaining a Grade C outcome. The outcome at 10 years of observation in patients attaining a Grade C result was significantly worse compared to patients attaining a Grade A (Chi-square test= 19.23, *p* < 0.001). The curves of actuarial patency rates and numbers of patients at risk are presented in Figure 4.

### 4.6. Factors Influencing Long-Term Outcome

The percentage of patients losing patency of anastomosis gradually increased over time in both groups. Cox proportional hazard regression analysis showed that an unsuccessful attempt at injury repair in a public surgical ward, complications overlapping the injury before the patient’s referral to the referential center, a multitude of therapeutic procedures before the patient’s referral, and need for advanced treatment preparing patients’ for repair, as well as postoperative complications after attaining primary or secondary patency of the anastomosis, were the factors contributing significantly to decreasing rates of actuarial patency. Differences due to patients’ sex and age, the type and severity of the injury, timing of patients’ referral, and timing of their preparation to repair were not statistically significant. The detailed results of the analysis are presented in Table 6.

## 5. Discussion

The new method of reporting outcomes of bile duct repair is based on the concept of “patency” of the bilo-jejunal anastomosis [20,21,22], unlike previous evaluations that were based mostly on clinical symptoms and biochemical tests [4,5,6,12,13,14,15,20,21,22,23,28,29,30,31,32,33,34]. However, it is stressed in most such reports that it is not the severity of the injury that is responsible for bad outcomes of repair. These are bad surgical behaviors, just as it happened to many of our patients, especially those of Group Ic, Group IIa, and Group IIb (as described in the Section 2), that caused results worse than they could.

Our study showed that primary patency of the damaged bile ducts could be attained in a significant number of cases provided that no attempts at repair injury are made prior to preparation for the relocation of patients from a public surgical ward to the specialized center. Notably, the timing of a patient’s transfer and the timing of preparing to repair also play a vital role in success [10,12,15,25,33,34,35,36,37,38,39,40]. None of the patients in Group I were transferred later than 6 weeks after injury (mean 25.4, SD +/−17.1). Preparing them for repair, even if the cases were very complicated and septic, did not exceed 5 weeks (mean 21.1, SD +/−15.4). These circumstances made it possible to properly classify precise diagnostics of side effects of injury and effectively prepare the patients for repair. From this point of view, our observations support the belief previously put forward by many researchers that the timing of the transfer is of crucial importance to achieving the intended effect of treatment [25,26,27,36,41,42,43,44]. Surely, the time should not exceed the minimum needed to establish the clinical status of the patient after index operation and to provide effective drainage of the peritoneal cavity, especially if the case seems to be ambiguous from the medical point of view [9,15,27,36,38,39,40,43,44]. So, primary patency of a biliary tree was possible to be achieved in 435 out of 442 patients of Group I relocated with injury without any first attempt to repair, except for seven who died due to cardiovascular complications in the post-operative period. The rate of attaining primary patency was assessed at 98%. The result corresponds to some other recently published studies [21,22,23,24,29,30,31,32,33]. Two hundred twenty-seven patients of Group II did not have this chance at all. They died not attaining primary patency because their repair failed, or they lost the patency due to a post-operative stricture that developed soon after the repair because of scar tissue and inflammatory changes at the site of anastomosis. These are usually difficult and neglected patients, transferred after weeks or months of unsuccessful surgical and/or endoscopy management with the hope of effective re-repair and regaining health [9,16,18,38,39,40,41,42,43]. Decision toward re-repair was taken with the belief that secondary patency could not be achieved and maintained by endoscopic, IR, or other treatment, even if stents or bouginage procedures will be retained up to 18 months, or longer, from the failure of primary repair. In other words, it was impossible to achieve a result of Grade B in any of the considered patients [20,21,22,30,31,32,33]. Preparing for surgery was challenging in most septic and complicated cases but did not exceed 5 weeks. Unfortunately, 18 patients (7.92%) of Group IIa died in the postoperative period. Of them, 42% died due to unsuccessful secondary repair and septic complications that developed in the postoperative period that needed further surgical and/or endoscopic interventions. Thus, secondary patency of bile tree was achieved in circa 92% of cases. Unexpectedly, the rates of patients attaining primary patency and secondary patency turned out to be very similar, despite clinical disparities between groups, and significant differences did not emerge until the second year of the later observation. The experience presented here is in line with some recent studies on this subject. Martínez-Mier et al. determined the percentage of primary patency at 93.4% by the initial surgical treatment [29,30]. In the study by Lindemann et al., primary and secondary patency were determined at 98.1% and 96.4%, respectively [22]. Rueda de Leon et al. [23] determined that re-repairs of duct injury have generally lower patency rates when compared with primary repairs. The differences were even more pronounced if the comparison was made between a middle-income country and a high-income country. In two other reports that were published in addition to the proposed standards, Cho et al. presented their results, where primary patency was attained in 94%, and Cuendis-Valazquez et al. reported even 100% attained primary patency rate, however, by using minimally invasive bile duct injury repair. [20,21,32]. Interestingly, Grade B or D results are not reported in surgical reports. Such results were reported by Koppatz et al., however, the study concerned endoscopy treatment of patients losing patency after the initial result of Grade A [40].

The outcome of bile duct injury repair in our patients was assessed by actuarial patency rates of anastomosis, according to proposed new standards [20,21,22]. There is a discussion on how the approach may be used for evaluating patients who underwent successfully primary repair and patients who have had prior failed attempts at repair. The study by Rueda De Leon et al. proved that standardized reporting outcomes after primary repair are applicable to re-repaired patients [23]. We adopted this point of view to compare such different clinical populations, as the patients undergoing primary and secondary bile duct repair. Our analysis revealed that the outcome of patients attaining a Grade A result was much better compared to patients attaining a Grade C result. Such tendency was present at any stage of observation, starting from the second year after repair. The number of patients losing patency of anastomosis gradually increased over time in all groups. The actuarial patency rates in patients of Group I were: 98%, 91%, and 83% in Group Ia, 96%, 90%, and 75% patients in Group Ib, 87%, 82%, and 60% patients in Group Ic, at 2, 5, and 10 years of follow-up, respectively. The actuarial patency rates in patients of Group II were: 86%, 83%, and 53% in patients of Group IIa, and 87%, 84%, and 55% in Group IIb, at 2, 5, and 10 years of follow-up, respectively. The results quoted here correspond with the patency rates published recently by authors who used new assessment standards. Originally Cho et al. presented their results, where primary patency was attained in 94%, and the 5- and 10-year Grade A result was 92% [20,21]. Cuendis-Velázquez et al. recently published outcomes after minimally invasive bile duct injury repair of Strasberg type E in Mexico also and reported 100% attained primary patency rate and a Grade A result in 93% of patients undergoing laparoscopic repair at a median 49-month follow-up and 100% in patients undergoing robotic repair at a median 16-month follow-up [32]. In the other study from Mexico, patency rates of 83% and 63% were reported in patients attaining a Grade A result at 10-year follow-up [29,30]. Lindemann et al. found primary and secondary patency at 98.1% and 96.4%, respectively, with no differences in 30-day complications [32]. Rueda de Leon et al. determined that re-repairs of bile duct injuries had lower actuarial patency rates when compared with primary repairs, and these differences were even more pronounced if the comparison was made between an upper-middle-income country and a high-income country [23]. Our research confirms this point of view and shows that a truly good repair result may be obtained in patients transferred from the public hospital early, with properly secured drainage of the peritoneal cavity and without infectious complications, as in patients of Group Ia and Ib. Therefore, patients of Group Ic who were transferred late and who were repaired delayed had the outcome more akin to patients attaining a Grade C result, although they attained a Grade A result of repair. The putative cause was undoubtedly intraabdominal infection and a multitude of therapeutic procedures before referral of a patient and the need for advanced treatment preparing patients for delayed repair. Some argue that the waiting time increases the complication rate, because of possible drainage obstruction or displacement, and that the deferred treatment is difficult to maintain in the outpatient setting. Kirks et al. suggested that they can perform the bile reconstruction within a median of 2 days after admission of the patient, resulting in an average length of stay of 11 days compared with a 32-day average reported by authors who defer the treatment [31,33]. In our opinion, such an approach to treatment may be effective in selected cases, i.e., in patients of our Group Ia but it is unreal and meaningless in patients included in Group IIa or IIb. Preparing them for repair was truly burdensome, since 18 required re-laparotomy, lavage, and drainage of the peritoneal cavity, 32 additionally USG guided drainage of peritoneal abscesses, 17 ERCP and stenting, as well as most of them TPI, general supplementation, and antibiotic therapy in according to bacteriologic seedings. This group of patients affected in a fundamental way the overall mortality within the population undergoing surgical treatment of bile duct injury because of the highest rate of postoperative complications and deaths.

Several factors could have an impact on actuarial patency rates. In the studies performed by Lindemann et al., they found that early referrals to their HPB center were predictors of loss of bile duct injury repair patency due to incomplete biliary tree imaging [22]. Their experience, however, remains in isolation. Most studies stress that actuarial primary patency rates and Grade A results are attained if the first repair was performed by a hepatobiliary surgeon. Certainly, an incomplete depiction of bile duct injury before reconstruction is one of the most important and could independently be associated with loss of anastomosis patency requiring revision. The re-repair increases the complexity of repeat biliary reconstruction in an HPB unit that affects distant results [22,23]. It is also stressed in many reports that patients who underwent a significantly higher number of surgical procedures, such as laparotomy plus drainage and/or bile duct biliary repair attempts prior to the referral to a third-level hospital are much more at risk of losing anastomosis patency when compared to patients repaired in specialized hepatobiliary centers. Gustavo Martinez-Mier et al. in the study from the Mexican Institute of Social Security, concerning factors contributing to bile duct repair failure, established primary patency at 93.4%, and ten-year actuarial patency at 53.9%. They showed that 90-day biliary complications and stenosis developing during the index treatment period were the factors that impacted the actuarial patency rate. They concluded that postoperative cholangitis is associated with loss of patency and had a potentially detrimental effect on the actuarial patency rate in bile duct injury repair [29,30]. The arguments put forward by the above-mentioned studies present, for the most part, the well-known causes of failure in biliary reconstruction stressed by many reports in the last twenty years [4,5,6,7,8,9,13,14,15,25,26,27,39,40,41,42,43]. The Cox proportional hazard regression model that was used in the present analysis showed that unsuccessful attempts of injury repair in the public surgical ward, complications overlapping the injury before the patient’s transfer to the referential center, a multitude of therapeutic procedures before patient referral, the need for advanced treatment preparing patients for repair, as well as postoperative complications after attaining primary or secondary patency of the anastomosis, were the factors contributing significantly to decreasing rates of actuarial patency.

The strengths of the current study are the large sample size and the long study period. Moreover, the analysis of repair outcomes has been matched to the new rating system proposed by international experts. However, the study is limited by its retrospective nature with all inherent limitations, as well as by the heterogeneity of the group of patients included in the study. The material was accumulated over the years according to the predetermined rules, but patients’ data and outcomes were analyzed on 31 December 2020. This required changes in many concepts which could lead to some distortions and bias. For example, different types of classifications of bile duct injury that were used over years in different surgical departments required standardization. Information concerning cholecystectomy, the circumstances of the incident and diagnosis of injury, as well as subsequent complications and modes of treatment applied before the patient’s referral may also be sources of bias since medical reports and interviews with the treatment teams do not fully reflect the management of the patients prior the referral. However, the grading of biliary injuries used for the analysis was based on these findings. No matter how well they have been verified, they may unintentionally have been biased. As this study addresses the major bile duct injury treated surgically, our analysis misses the whole population of patients treated endoscopically with success, except those who needed temporally stenting or endoscopic dilatation of the bile tree aiming to prepare the patient for surgical re-repair. This retrospective study also omits QOL measurement and of patients Grades B–D who subsequently lose patency but could be brought or restored to secondary patency by surgery or endoscopic treatment. The analysis to this extent exceeded the scope of this study and deserves an extensive discussion of its own. It should be noted, however, that all procedures of bile duct injury repair by hepato-jejunostomy were performed in accordance to art, by the same highly experienced specialists. Nevertheless, the cause-effect relationship between procedures made, and the evaluation of a patient long-term outcome may have given rise to a small amount of bias in the interpretation of the results. It must be underlined that this current study is biased by the fact that it was performed at a referral center for hepatobiliary surgery and may therefore not reflect how iatrogenic bile duct injuries are managed at the level of public surgical wards. We believe, however, that the results of this study can be generalized due to a large number of patients included and followed up for a relatively long period of time. Most of these patients are still being monitored, including some of those who underwent repair in the 1990s.

The role of endoscopic procedures in the patients included in the study was subsidiary. ERCP was routinely used for diagnostic purposes, however, most of the patients from Group Ic, IIa, and IIb had permanent plastic stents re-placement, either for treatment purposes of biliary fistula (patients from Group Ic, IIa) or cholangitis due to biliary stricture (patients of Group IIb). Decision toward surgery in all these cases was taken with the belief that the bile ducts patency could not be achieved and maintained by endoscopic procedures, even if stents or bouginage procedures will be retained for up to 18 months or longer [17,18,19,29,30].

## 6. Conclusions

-Patients with bile duct injury after cholecystectomy should be referred immediately or without undue delay to a tertiary hepatobiliary unit for repair.-Patients from public surgical wards who did not attain primary patency due to failure in attempts at repair, or patients who early lost patency because of stricture developing in faulty performed anastomosis could be effectively re-repaired attaining a repair of Grade C.-Late transfer, ineffective drainage of the peritoneal cavity, infectious complications, and a multitude of therapeutic procedures before referral and the need for advanced treatment preparing to repair worsens actuarial patency rates even if the patients attain a Grade A repair.-Re-repair increased the complexity of biliary reconstruction, however, the rates of patients attaining primary patency and secondary patency were very similar to the index time of treatment.-The significant differences in actuarial patency rates between patients attaining Grade A and Grade C results did not emerge until 2 years, and at 10 years became worse in the patients attaining a repair result of Grade C.-New proposed standards for outcome reporting allowed the comparison of such different populations from the clinical point of view.

## Figures and Tables

**Figure 1 jcm-11-03396-f001:**
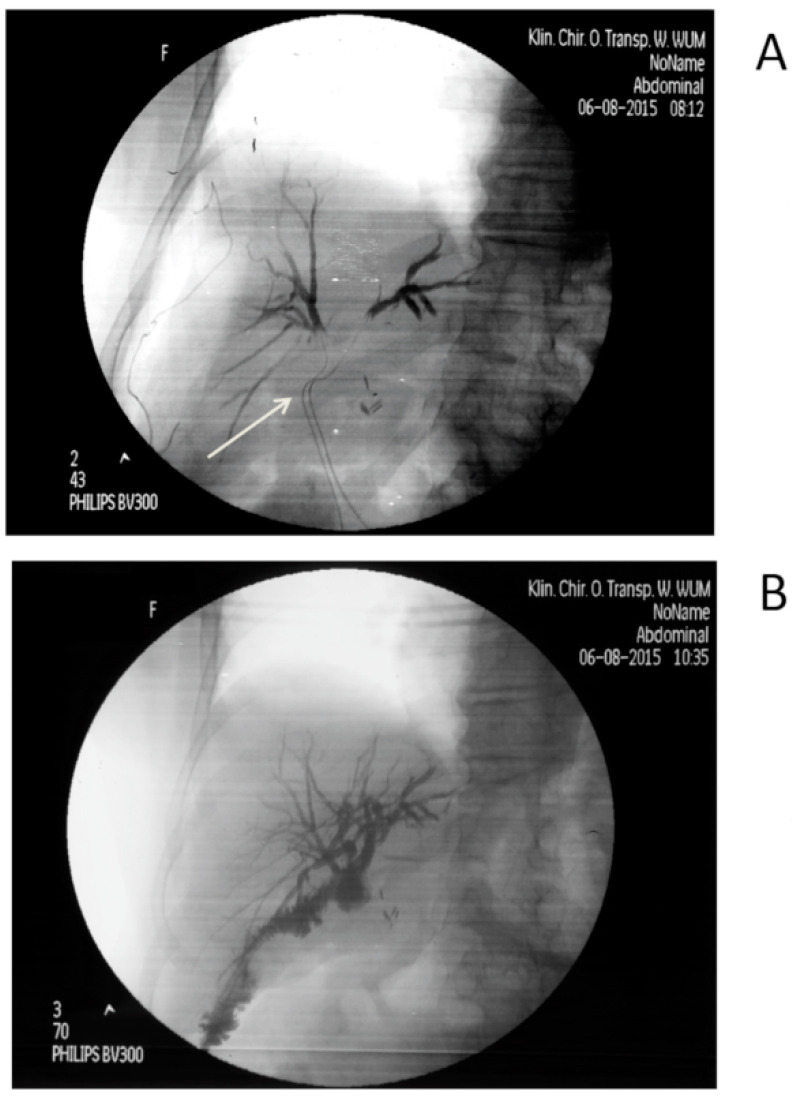
(**A**) Cholangiogram by intrahepatic catheters inserted after the stump of hepatic duct was opened (see white arrow) aiming to visualize whether all intrahepatic branches will be drained through the anastomosis. (**B**) Cholangiogram by intrahepatic catheters inserted after the bilo-jejunal anastomosis was completed aiming to visualize the tightness of the anastomosis and easy outflow of contrast medium to jejunum.

**Figure 2 jcm-11-03396-f002:**
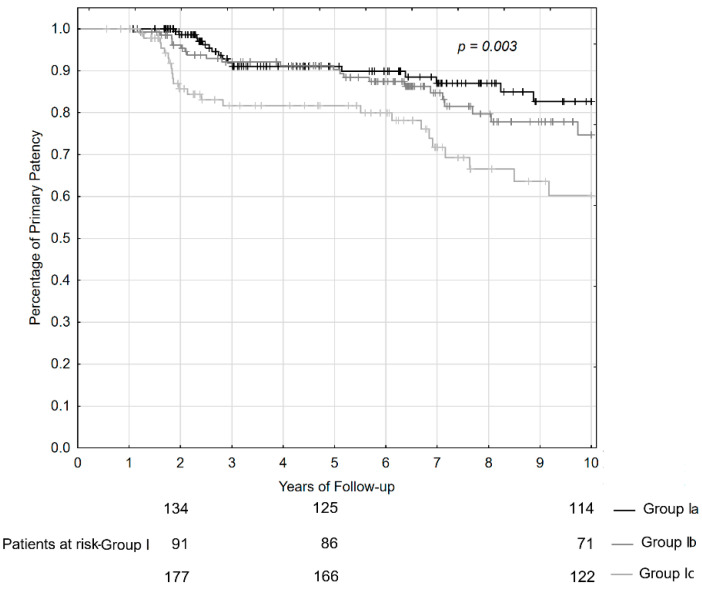
Primary patency curves and numbers of the patients who attained Grade A outcomes of repair at the period of 2, 5, and 10 years of follow-up.

**Figure 3 jcm-11-03396-f003:**
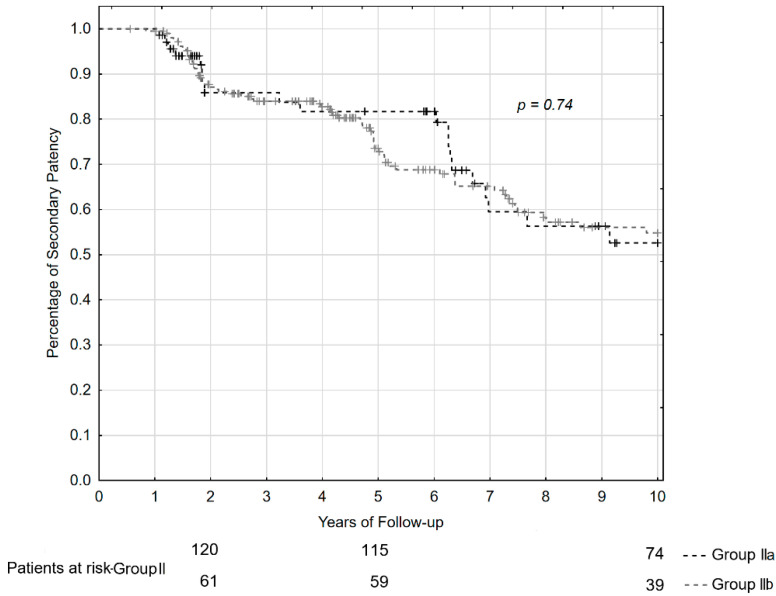
Secondary patency curves and numbers of the patients who attained a Grade C outcome of repair at the period of 2, 5, and 10 years of follow-up.

**Figure 4 jcm-11-03396-f004:**
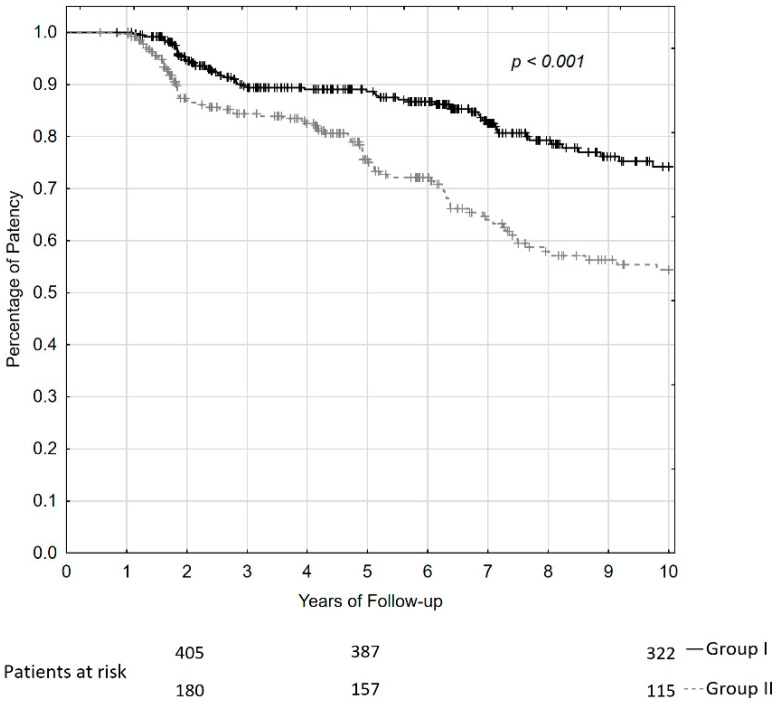
The patency curves and numbers of the patients who attained Grade A and Grade C outcomes of repair at the period of 2, 5, and 10 years of follow-up.

**Table 1 jcm-11-03396-t001:** Clinical details of patients constituting Group I for whom no repair of injury has been attempted at public surgical wards.

Patients	Type of Injury (Acc. to Strasberg)	Grade of Injury Severity	Pattern of Referral
Group Ia 137 pts.	D–31 E1/E3–72 E4–34	Grade 1–31 Grade 2–72 Grade 3–34	Effective peritoneal drainage done intraoperatively. Referred for repair.
Group Ib 95 pts.	E1/E3–95	Grade 2–95	Additional drainage of the peritoneal cavity was done by USG prior to the referral. Referred for repair.
Group Ic 210 pts.	D–47 E1/E3–103 E–4–60	Grade 1–47 Grade 2–103 Grade 3–60	Bile fistula, ineffective peritoneal drainage. Intra-abdominal and general infection. Before the transfer, all the patients underwent wide spectrum antibiotic therapy, and: - 55 patients underwent re-laparotomy, lavage, and drainage of the peritoneal cavity, - 47 underwent additional percutaneous USG-guided drainage of peritoneal biloma, - 87 underwent endoscopic stenting for bile fistula Referred for advanced treatment and repair after proper preparation.

**Table 2 jcm-11-03396-t002:** Clinical details of patients constituting Group II who developed complications due to the failure in attempts to repair the injury at public surgical wards.

Patients	Type of Injury (Acc. to Strasberg)	Grade of Injury Severity	Pattern of Referral
Group IIa 157 pts.	D–31 E1/E3–72 E4–83	Grade 2–72 Grade 3–83	Separation of biliary-jejunal anastomosis. Bile leak and external biliary fistula. Abdominal infection. Before the transfer, all the patients underwent wide spectrum antibiotic therapy, and: - 64 patients underwent re-laparotomy, lavage, and drainage of the peritoneal cavity. - 60 underwent percutaneous USG-guided drainage of peritoneal biloma. - 107 underwent endoscopic stenting for bile leak. - Referred for advanced treatment and re-repair.
Group IIb 70 pts.	D–31 E1/E3–25 E4–14	Grade 1–31 Grade 2–25 Grade 3–14	Early stricture of biliary-jejunal anastomosis. Recurrent episodes of cholangitis. Before the transfer, all the patients underwent a wide spectrum of antibiotic therapy, and repeatedly endoscopic biliary dilatation and prosthesis procedures. Referred for re-repair.

**Table 3 jcm-11-03396-t003:** Clinical condition and the scope of preparing the patients for biliary repair.

Patients		Clinical Condition	Scope of Preparing
Group I	Ia—137 pts.	Good condition—137	Improvement of general status. Continuation of antibiotic therapy
	Ib—95 pts.	Good condition—95	As above + control of sepsis and drainage from the peritoneal cavity. Antibiotic therapy is according to bacteriologic seedings. Additional percutaneous USG guided drainage of peritoneal biloma in 26 patients
	Ic—210 pts.	Good condition—21 Average condition—104Poor condition—87	As above + additional percutaneous USG guided drainage of peritoneal biloma in 35 patients, endoscopic stenting for bile fistula in 58 patients, TPI in 21 patients
Group II	IIa—157 pts.	Average condition—132 Poor condition—25	As above + re-laparotomy, lavage, and drainage of the peritoneal cavity in 18 patients, additional USG guided drainage of peritoneal biloma in 32 patients, endoscopic stenting in 17 patients, TPI in 43
	IIb—70 pts.	Average condition—70	General supplementation. Antibiotic therapy according to bacteriologic seedings, endoscopic dilation of the bile ducts and stenting in 16 patients, TPI in 5 patients

**Table 4 jcm-11-03396-t004:** Results of repair and postoperative complications in 669 patients with bile duct injury.

Item		Number of Patients
Repair surgery by hepatico-jejunostomy		669
Postoperative course uncomplicated		534 (79.9%)
Postoperative course complicated		135 (20.1%)
Grade 1: Postoperative pain, bile leak around stents, wound infection		44
Grade 2: Postoperative anemia, blood transfusion, total parenteral nutrition		31
Grade 3: Leak from anastomosis, bile collection requiring drainage		18
Grade 4: Sepsis, pneumonia, organ insufficiency, thromboembolic disease		17
Grade 5: Multiorgan failure, patient’s death:	patients of group Ia, Ib, IIb	none
	patients of group Ic	7 (1.0%)
	patients of group IIa	18 (2.7%)
Overall mortality		25 (3.7%)

**Table 5 jcm-11-03396-t005:** Clinical differences in studied groups of patients with bile duct injuries.

Patients		Timing of Referral ((Mean Days)	Timing of Preparation to Repair (Mean Days)	Postoperative Death	Rate of Re-Admission	Number of Patients in Follow-Up	Result of Repair
Group I	Ia—137 pts.	5.2 (+/−2.1)	8.5 (+/−5.9)	0	0	137	Grade A
	Ib—95 pts.	11.7 (+/−5.8)	15.6 (+/−7.1)	0	0	95	
	Ic—210 pts.	25.4 (+/−17.1)	21.1 (+/−15.4)	7 (3%)	0	203	
Group II	IIa—157 pts.	127.0 (+/−38.2)	23.8 (+/−12.1)	18 (11%)	0	139	Grade C
	IIb—70 pts.	263.8 (+/−52.1)	24.3 (+/−11.4)	0	0	70	

**Table 6 jcm-11-03396-t006:** Results of Cox proportional hazard regression.

Parameter	Hazard Ratio (95% CI)	Statistic Value (Chi-Square)	*p*-Value
Attempt at injury repair at public wards	2.091	6.9769	0.008
Complications overlapping the injury before the patient’s referral to a referential center	2.850	30.7684	<0.0001
A multitude of therapeutic procedures before a patient’s referral	2.379	5.1330	0.0235
Need for advanced treatment preparing patients to repair	3.072	12.1902	0.0005
Postoperative complications after attaining primary or secondary patency of the anastomosis	2.309	6.0725	0.0137

## Data Availability

Not applicable.

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
