# Peer review of "Actuarial Patency Rates of Hepatico-Jejunal Anastomosis after Repair of Bile Duct Injury at a Reference Center"

_jcm, 2022, doi:10.3390/jcm11123396_

Round 1
Reviewer 1 Report
The authors submitted a revised version of manuscript entitled: ‘’ +Actuarial patency rates of hepatico - jejunal anastomosis after repair of bile duct injury at a reference center’’. They evaluated outcomes of 669 patients with bile duct injuries Strasberg D and E type referred to the department from public surgical wards between 1990 and 2020.
They reported that twenty-five (3.7%) patients died after repair surgery. Actuarial patency rates at 2, 5, and 10 years of follow-up were 93%, 88% and 74% or 86%, 75%, and 55% in patients attaining of Grade A and Grade C outcome respectively. The results of their study clearly showed that bile duct injury stands out as a surgical challenge, requiring specialized management at a referral center. Improper proceeding after injury is the factor leading to faster loss of anastomotic patency.
The study is well-designed and written and of great importance for laparoscopic surgeons dealing with biliary pathology. I would like to take the opportunity to congratulate the authors on a well-written study.
In the previous revision I have addressed several issues that needed to be revised. The authors adequately responded to all of my remarks and in my opinion this study may be accepted for publication in present form.
Reviewer 2 Report
No more comments
This manuscript is a resubmission of an earlier submission. The following is a list of the peer review reports and author responses from that submission.
Round 1
Reviewer 1 Report
The authors evaluated outcomes of 669 patients with bile duct injuries Strasberg D and E type referred to the department from public surgical wards between 1990 and 2020.
The study is well-designed and of great importance for laparoscopic surgeons. Congratulations to the authors on a well-written study. However, several, mostly minor issues need to be improved before any favorable decision should be made.
My concerns are as follows:
- Introduction. It is very important to highlight the incidence of CBD injuries. Please add following statements and references: a) ‘’The incidence of iatrogenic main bile duct lesions has significantly increased, with incidences ranging from 0.2 to 1.5% according to current studies. REF: Systematic review of the role of indocyanine green near-infrared fluorescence in safe laparoscopic cholecystectomy (Review). Exp Ther Med. 2022;23(2):187. doi: 10.3892/etm.2021.11110. b) An incidence of bile duct injuries in the pediatric population is slightly higher than in adults probably due to the less experience of pediatric surgeons with laparoscopic cholecystectomy REF: Gallbladder Disease in Children: A 20-year Single-center Experience. Indian Pediatr. 2019;56(5):384-386. c) Also, an incidence of CBD injuries is greater in regional hospitals than in University hospitals or larger centers. REF: Laparoscopic cholecystectomy in Cantonal Hospital Livno, Bosnia and Herzegovina and University Hospital Center Split, Croatia. Coll Antropol. 2010;34:125-8.
- Methodology – Please indicate primary and secondary outcomes of the study in a separate paragraph in methodology.
- Methodology – Please indicate clear inclusion and exclusion criteria for the study.Whether the pediatric patients were included in this analysis, if yes what was the percentage of pediatric patients?
- Statistical analysis – Please indicate which statistical test was used to test normality of distribution of the data.
- Statistical analysis / results – ‘’Chi^2’’ – Please replace with Chi-square test
- Please replace Ethical approval before Statistical analysis chapter, it should be part of methodology and add the IRB reference and date of approval.
- Table 5 – Postoperative death (please indicate what the numbers represent) n? if yes please add percentage in brackets. Also ‘not applicable’’ that means zero or? If that was zero please state as zero.
- Conclusions should be summarized and shortened. Do not present general statements or personal impressions. In conclusion, state only conclusions (finding) from your study results.
Author Response
WÅ‚odzimierz Otto, Professor of Surgery, Md, PhD
Medical University of Warsaw,
02-097 Warsaw, Banacha 1a, Poland
Answers to The Reviewers
Introduction. It is very important to highlight the incidence of CBD injuries. Please add following statements and references: a) ‘’The incidence of iatrogenic main bile duct lesions has significantly increased, with incidences ranging from 0.2 to 1.5% according to current studies. REF: Systematic review of the role of indocyanine green near-infrared fluorescence in safe laparoscopic cholecystectomy (Review). Exp Ther Med. 2022;23(2):187. doi: 10.3892/etm.2021.11110. b) An incidence of bile duct injuries in the pediatric population is slightly higher than in adults probably due to the less experience of pediatric surgeons with laparoscopic cholecystectomy REF: c) Also, an incidence of CBD injuries is greater in regional hospitals than in University hospitals or larger centers. REF: Laparoscopic cholecystectomy in Cantonal Hospital Livno, Bosnia and Herzegovina and University Hospital Center Split, Croatia. Coll Antropol. 2010;34:125-8.
Answer: Your suggestion are in the text
Methodology – Please indicate primary and secondary outcomes of the study in a separate paragraph in methodology.
Answer: It’s done, see text
Methodology – Please indicate clear inclusion and exclusion criteria for the study. Whether the pediatric patients were included in this analysis, if yes what was the percentage of pediatric patients?
Answer: The criteria were indicated, There were no pediatric patients around, since the department is only for adult patients (over 18),
Statistical analysis – Please indicate which statistical test was used to test normality of distribution of the data.
Answer: The Shapiro-Wilk test was used to assess the normality distributions of the study variables.
Statistical analysis / results – ‘’Chi^2’’ – Please replace with Chi-square test
Answer: Changes have been done, see text
Please replace Ethical approval before Statistical analysis chapter, it should be part of methodology and add the IRB reference and date of approval.
Answer: Done
Table 5 – Postoperative death (please indicate what the numbers represent) n? if yes please add percentage in brackets. Also ‘not applicable’’ that means zero or? If that was zero please state as zero.
Answer: done, see text
Conclusions should be summarized and shortened. Do not present general statements or personal impressions. In conclusion, state only conclusions (finding) from your study results.
Answer: changed according to your suggestion
I would like to thank The Reviewers for their time and suggestions.
With regards
WÅ‚odzimierz Otto
Reviewer 2 Report
Comments on: Actuarial patency rates of hepatico-jejunal anastomosis after repair of bile duct injury at a reference center
The authors have analyzed a large cohort of patients employing new proposed standards for outcome reporting after bile duct repair following injury during laparoscopic cholecystectomy. They clearly showed that primary repair produced better results as compared to secondary repair when initial attempts have failed. More to that, differences emerged at prolonged follow-up. However, I have comments that have to be discussed prior to considering the article for acceptance.
The grouping of patients is too complicated and disturbing, similar patients being assigned to multiple different groups. When reading the article this becomes a real challenge not to be lost among those 5 groups. It is known that the difference in results of primary repair depends mainly on the presence or absence of an intraabdominal abscess. The same is true for a secondary repair. Details of preparation of patients for repair seems excessive. Abundance of various details, having no real influence on final result makes it even more complicated.
On the contrary, surgical technique is described superficially. Were the intraluminal drains routinely used and why, which anastomotic technique was applied, have the tailoring of the hepaticojejunal anastomosis not changed during this very long study period, was Hepp-Couinaud technique utilized, etc.
Strangely the results seem not to be influenced by severity of injury which was in detail presented in Methods section, at least this issue was not analyzed. No data on concomitant vascular injuries is presented.
Unfortunately, the presented manuscript is very lengthy, with numerous “essays”, with numerous repetitions, especially the Discussion section.
The same has to be said about conclusions, which are merely continuation of discussion with the main message deeply concealed among the excessive wording.
Summarizing the authors have collected a huge piece of valuable data; however, this has to be presented in a more concise manner, as a reader seeks for a brevity of text, clarity of presentation and condensed conclusions.
Author Response
WÅ‚odzimierz Otto, Professor of Surgery, Md, PhD
Medical University of Warsaw,
02-097 Warsaw, Banacha 1a, Poland
Answers to The Reviewers
The grouping of patients is too complicated and disturbing, similar patients being assigned to multiple different groups. When reading the article this becomes a real challenge not to be lost among those 5 groups. It is known that the difference in results of primary repair depends mainly on the presence or absence of an intraabdominal abscess. The same is true for a secondary repair. Details of preparation of patients for repair seems excessive. Abundance of various details, having no real influence on final result makes it even more complicated.
Answer: I’m sorry about that. Grouping of patients has been changed according to your suggestion. In fact, we analyzed only two groups of patients – these who had not attempts to repair the injury by public surgeons constituted GROUP I, with subgroups Ia, Ib, Ic indicating clinical differences between them, and those who had attempts to repair that failed GROUP II, with subgroups IIa and IIb indicating clinical differences between them. Indication of clinical details are the key to show why the outcome after the repair is different in these who attained primary or secondary patency.
On the contrary, surgical technique is described superficially. Were the intraluminal drains routinely used and why, which anastomotic technique was applied, have the tailoring of the hepaticojejunal anastomosis not changed during this very long study period, was Hepp-Couinaud technique utilized, etc.
Answer: Surgical technique was not a scope of the study since the roles of execution of hepatico-jejunostomy are well established and generally known. During the time, there were operations made by the same surgeons, in the same conditions, the same roles of performance, the same type of suteres , etc,,,. This what has changed was anesthesia and ICU cure.
Strangely the results seem not to be influenced by severity of injury which was in detail presented in Methods section, at least this issue was not analyzed. No data on concomitant vascular injuries is presented.
Answer: Certainly, it was not the severity of injury cause the differences in outcome. All patients had Strasberg D/E type injuries. These were bad surgical behaviors applied especially to patients of group Ic and group IIa and IIb, which were described in Material/Method section, a reason that 40% of injured patients achieved just outcome of Grade C, and their actuarial patency rates were lower than they could be.
Unfortunately, the presented manuscript is very lengthy, with numerous “essays”, with numerous repetitions, especially the Discussion section.
Answer: it was shortened, the discussion from 3079 words to 2480 words.
The same has to be said about conclusions, which are merely continuation of discussion with the main message deeply concealed among the excessive wording.
Answer: conclusions were changed, see text
Summarizing the authors have collected a huge piece of valuable data; however, this has to be presented in a more concise manner, as a reader seeks for a brevity of text, clarity of presentation and condensed conclusions.
Answer: I hope that these changes made to the manuscript will improve the understanding of the research topic undertaken in our work.
I would like to thank The Reviewers for their time and suggestions.
With regards
WÅ‚odzimierz Otto
Reviewer 3 Report
Well written study on an interesting topic. In order to improve the quality of the manuscript I would suggest 3 minor corrections
1) Please shorten the Discussion. It is too big and it is tiring for the reader.
2) Add a small paragraph commenting on the use of plastic stents (one, two or three) placed on the common bile duct during ERCP in order to dilate the bile duct and restore patency
3) I would add in the limitations of the study the heterogeneity of the group of patients included in the study
Author Response
WÅ‚odzimierz Otto, Professor of Surgery, Md, PhD
Medical University of Warsaw,
02-097 Warsaw, Banacha 1a, Poland
Answers to The Reviewers
- Please shorten the Discussion. It is too big and it is tiring for the reader.
Answer: discussion was shortened from 3079 words to 2480 words.
- Add a small paragraph commenting on the use of plastic stents (one, two or three) placed on the common bile duct during ERCP in order to dilate the bile duct and restore patency
Answer: done, see text
- I would add in the limitations of the study the heterogeneity of the group of patients included in the study
Answer: done, see text in part about study limitations.
I would like to thank The Reviewers for their time and suggestions.
With regards
WÅ‚odzimierz Otto
Round 2
Reviewer 2 Report
The revision is rather formal.
The conclusions remain not based on the results of the study.